



**Responses of elemental content and macromolecule of the coccolithophore**
***Emiliania huxleyi* to reduced phosphorus availability and ocean acidification**
**depend on light intensity**
**Yong Zhang[1,*], Yong Zhang[1,§], Shuai Ma[1], Hanbing Chen[2], Jiabing Li[1], Zhengke**
**Li[3], Kui Xu[4], Ruiping Huang[5], Hong Zhang[1], Yonghe Han[1], Jun Sun[6]**
[1]College of Environmental and Resource Sciences, College of Carbon Neutral
Modern Industry, Fujian Key Laboratory of Pollution Control and Resource
Recycling, Fujian Normal University, Fuzhou, China
[2]College of Life Science, Fujian Normal University, Fuzhou, China
[3]School of Food and Biological Engineering, Shanxi University of Science and
Technology, Xi'an, China
[4]Hubei Key Laboratory of Edible Wild Plants Conservation and Utilization, Hubei
Engineering Research Center of Special Wild Vegetables Breeding and
Comprehensive Utilization Technology, College of Life Sciences, Hubei Normal
University, Huangshi, China
[5]State Key Laboratory of Marine Environmental Science, College of Ocean and Earth
Sciences, Xiamen University, Xiamen, China
[6]Institute for Advanced Marine Research, China University of Geosciences,
Guangzhou, China



Running head: Physiology and biochemistry of *E. huxleyi*
*Correspondence: Yong Zhang (yongzhang@fjnu.edu.cn)
Keywords: Carbohydrate; $CO_2$; coccolithophore; elemental content; light intensity;
phosphorus availability; protein.
§Email: qsx20211022@student.fjnu.edu.cn












**Abstract**


Global climate change leads to simultaneous changes in multiple environmental
drivers in the marine realm. Although physiological characterization of
coccolithophores have been studied under climate change, there is limited knowledge
on the biochemical responses of this biogeochemically important phytoplankton
group to changing multiple environmental drivers. Here we investigate the interactive
effects of reduced phosphorus availability (4 to 0.4 μmol L$^{-1}$), elevated $p$CO$_2$
concentrations (426 to 946 μatm) and increasing light intensity (40 to 300 μmol
photons m$^{-2}$ s$^{-1}$) on elemental content and macromolecules of the cosmopolitan
coccolithophore *Emiliania huxleyi*. Reduced phosphorus availability reduces
particulate organic nitrogen and protein contents under low light intensity, but not
under high light intensity. Reduced phosphorus availability and ocean acidification
act synergistically to increase particulate organic carbon (POC) and carbohydrate
contents under high light intensity but not under low light intensity. Reduced
phosphorus availability, ocean acidification and increasing light intensity act
synergistically to increase the allocation of POC to carbohydrates. Under future ocean
acidification and increasing light intensity, enhanced carbon fixation could increase
carbon storage in the phosphorus-limited regions of the oceans where *E. huxleyi*
dominates the phytoplankton assemblages. In each light intensity, elemental carbon to
phosphorus (C : P) and nitrogen to phosphorus (N : P) ratios decrease with increasing
growth rate. These results suggest that coccolithophores could reallocate chemical
elements and energy to synthesize macromolecules efficiently, which allows them to
regulate its elemental content and growth rate to acclimate to changing environmental
conditions.

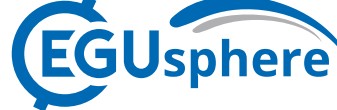



## 1 Introduction

Continuous increase in atmospheric $CO_2$ level, as a consequence of anthropogenic activities, leads to global and ocean warming, which in turn shoals the ocean upper mixed layer (UML), hinders upward transport of nutrients from deeper ocean to the UML and increases light exposures to phytoplankton cells dwelling therein (Steinacher et al., 2010; Wang et al., 2015). The dissolution of $CO_2$ in the oceans is causing a significant chemical shift toward higher $CO_2$ and proton ($[H^+]$) concentrations, a process defined as ocean acidification (OA) (Caldeira and Wickett, 2003). These ocean changes expose phytoplankton cells within the UML to multiple drivers, and understanding the effects of changing multiple environmental drivers on the physiology and biochemistry of marine phytoplankton is important for projections of changes in the biogeochemical roles of phytoplankton in the future ocean (Gao et al., 2019).

Coccolithophores take up carbon dioxide ($CO_2$) to produce particulate organic carbon (POC) via photosynthesis, and use bicarbonate ($HCO_3^-$) and calcium ($Ca^{2+}$) to synthesize calcium carbonate plates (coccoliths, PIC) and release $CO_2$ via calcification, and play a critical role in the marine carbon cycle (Rost and Riebesell, 2004). The cosmopolitan coccolithophore *Emiliania huxleyi* typically forms extensive blooms that are easily detected by satellite remote sensing due to high light scattering caused by coccoliths (Terrats et al., 2020; He et al., 2022). Within *E. huxleyi* blooms in polar and subpolar oceans, dissolved nitrate and phosphate concentrations in surface seawater could be as lower as 0.95 µmol $L^{-1}$ and 0.16 µmol $L^{-1}$, respectively (Townsend et al., 1994), light intensity are higher than 300 µmol photons $m^{-2}$ $s^{-1}$ (Tyrrell and Merico, 2004), and the mean concentrations of seawater $CO_2$ increased by 21.0%–43.3% which weakens the oceanic $CO_2$ uptake from the atmosphere



(Kondrik et al., 2018). *Emiliania huxleyi* is also the dominant phytoplankton species
in the lower photic zone in the north-eastern Caribbean Sea (western Atlantic Ocean)
(Jordan and Winter, 2000) and in the South Pacific Gyre where dissolved nitrate and
phosphate concentration are about 1.0 μmol L$^{-1}$ and 0.2 μmol L$^{-1}$, respectively, and
light intensity is lower than 20 μmol photons m$^{-2}$ s$^{-1}$ (Beaufort et al., 2008; Perrin et
al., 2016). To explore how *E. huxleyi* acclimate to simultaneous changes in
macronutrient concentration, light intensity and $CO_2$ level, it is interesting to
investigate their physiological and biochemical processes, which can also help to
project the effect of coccolithophores on ocean carbon cycle and ecological systems.
For more than a decade, research has shown that *E. huxleyi* cells developed several
strategies to acclimate to reduced phosphorus availability, increasing light intensity
and ocean acidification (Leonardos and Geider, 2005; McKew et al., 2015; Wang et
al., 2022). Interactive effects of phosphorus availability and light intensity have
shown that under phosphorus limitation condition, cells increased expression and the
activity of alkaline phosphatase, and took up and used phosphorus efficiently under
high light intensity, whereas they lowered the phosphorus uptake rate under low light
intensity (Riegman et al., 2000; Perrin et al., 2016). In addition, the positive effect of
reduced phosphorus availability on POC and PIC contents of *E. huxleyi* was further
enhanced by increasing light intensity due to high-light-induced increases in $CO_2$ and
$HCO_3^-$ uptake rates under low phosphate availability (Leonardos and Geider, 2005).
The negative effect of reduced phosphorus availability on particulate organic
phosphorus (POP) content was partly compensated by increased $PO_4^{3-}$ uptake rate
under increasing light intensity (Perrin et al., 2016). On the other hand, several studies
report that ocean acidification and reduced phosphorus availability acted
synergistically to increase the POC content, especially at high light intensity, and





acted antagonistically to affect PIC content of *E. huxleyi* (Leonardos and Geider, 2005;
Matthiessen et al., 2012; Zhang et al., 2020). In addition, ocean acidification normally
amplified the positive effect of increasing light intensity on POC content (Rokitta and
Rost, 2012; Heidenreich et al., 2019). Due to high proton concentration-induced
reduction in $HCO_3^-$ uptake rate, ocean acidification could weaken or counteract the
positive effect of increasing light intensity on PIC content (Rokitta and Rost, 2012;
Kottmeier et al., 2016). Overall, while recent studies have focused on physiological
performance of *E. huxleyi* and their effects on marine biogeochemical cycling of
carbon, little information is available about the biochemical response of *E. huxleyi* to
reduced phosphorus availability, increasing light intensity and ocean acidification.
The objective of this study is to investigate the combined effects of reduced
phosphorus availability, increasing light intensity and ocean acidification on cellular
elemental contents, the carbon (C) : nitrogen (N) : phosphorus (P) ratio and
macromolecules of *E. huxleyi*, and to analyze the effects of macromolecules on
elemental contents. Under reduced phosphorus availability, increasing light intensity
and ocean acidification, we hypothesize that increased POC content is more likely to
be caused by increased carbohydrate content. In addition, we discuss the potential
mechanisms for changing PIC content in response to changed levels of phosphate,
light, and $CO_2$, which is important for projections of changes in coccolithophore
biogeochemistry and ecology in the future ocean.

**2 Materials and Methods**
**2.1 Experimental setup**
*Emiliania huxleyi* strain RCC1266 (morphotype A) was originally isolated from shelf
waters around Ireland (49°30' N, 10°30' W) and obtained from the Roscoff algal

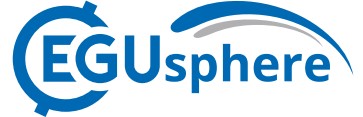

culture collection. *Emiliania huxleyi* was cultured under a 14 h : 10 h light : dark
cycle (light period: 06:00 to 20:00 h) in a thermo-controlled incubator (MGC-400H,
Shanghai Yiheng Scientific Instrument) at 20ºC in semicontinuous cultures. The
artifical seawater (ASW) media was prepared according to Berges et al. (2001) with
the addition of 2350 µmol $L^{-1}$ bicarbonate to achieve the total alkalinity (TA) of 2350
µmol $L^{-1}$, and enriched with 64 µmol $L^{-1}$ $NO_3^-$, f/8 concentrations for trace metals and
vitamins (Guillard and Ryther, 1962). The experiment was conducted in two parts
(Fig. S1). The first part (Part 1) was performed at 40 µmol photons $m^{-2}$ $s^{-1}$ (low light
intensity, LL) and the second one (Part 2) was at 300 µmol photons $m^{-2}$ $s^{-1}$ (high light
intensity, HL). The LL intensity used here corresponds to the lower end of the
irradiance range of the UML, and the HL intensity represents to the irradiance in the
surface ocean (Jin et al., 2016; Perrin et al., 2016). For each part of the experiment,
dissolved inorganic phosphorus (DIP) concentration and ocean acidification were
combined in a fully factorial design: high DIP concentration (4 µmol $L^{-1}$) ＋ low $CO_2$
level (426 µatm, current $CO_2$ level) (HP+LC, treatment 1 in LL and treatment 5 in
HL), high DIP concentration (4 µmol $L^{-1}$) ＋ high $CO_2$ level (946 µatm, future $CO_2$
level) (HP+HC, treatment 2 in LL and treatment 6 in HL), low DIP concentration
(0.43 µmol $L^{-1}$) ＋ low $CO_2$ level (426 µatm) (LP+LC, treatment 3 in LL and treatment
7 in HL), and low DIP concentration (0.43 µmol $L^{-1}$) ＋ high $CO_2$ level (946 µatm)
(LP+HC, treatment 4 in LL and treatment 8 in HL). High DIP concentration is replete
for physiological process of *E. huxleyi*, and low DIP concentration corresponds to the
upper end of the range of phosphate concentration in the coastal waters (Larsen et al.,
2004). There were eight treatments totally and four biological replicates for each
treatment (Fig. S1). In all cases, cell densities were lower than 78,000 cells $mL^{-1}$ and
the cells were acclimated to each treatment for at least 8 generations before



physiological and biochemical parameters were measured.

At LL intensity (Part 1), for the treatments of HP+LC and HP+HC, the ASW media

were enriched with 4 μmol $L^{-1}$ $PO_4^{3-}$ and aerated for 24 h at 20°C with filter-sterilized
(PTFE filter, 0.22 μm pore size, Nantong) air pumped from the room. The $pH_{Total}$
(total scale) values of the media under both HP+LC and HP+HC treatments were
about 8.04. The dry air was humidified with Milli-Q water prior to the aeration to
minimize evaporation. Under the HP+HC treatment, the $pH_{Total}$ values of the media
were adjusted to 7.74 by stepwise additions of $CO_2$-saturated seawater, and the ratio
was about 6.5 mL $CO_2$-saturated seawater : 1000 mL ASW media. The $CO_2$-saturated
seawater was achieved by bubbling pure $CO_2$ gas into 500 ml ASW media with a total
alkalinity of 2350 μmol $L^{-1}$ for 2 h. For the treatments of LP+LC and LP+HC, the
ASW media were enriched with 0.4 μmol $L^{-1}$ $PO_4^{3-}$ and aerated for 24 h at 20°C with
filtered room air. Under the LP+HC treatment, the $pH_{Total}$ values of the media were
also adjusted to 7.74 as described above. The HP+LC, HP+HC, LP+LC and LP+HC
seawater were then filtered (0.22 μm pore size, Polycap 75 AS, Whatman) and
carefully pumped into autoclaved 50 mL (for TA measurements), 600 mL (for
pre-experimental cultures) and 2350 mL (for experimental cultures) polycarbonate
bottles (Nalgene) with no headspace to minimize gas exchange. The volumes of
culture inoculum were calculated to match the volumes of media taken out from the
bottles prior to inoculation. The cells were inoculated to achieve an initial density of
5000 cells $ml^{-1}$ in the HP+LC and HP+HC conditions, respectively, and cultured for 2
days, then diluted to the initial density again. These processes were performed three
times in 600 mL bottles for pre-experimental cultures at 40 μmol photons $m^{-2}$ $s^{-1}$ (LL)
of photosynthetically active radiation (PAR; measured using a LI-190SA quantum
sensor, Beijing Ligaotai Technology Co. Ltd.). In the main experimental cultures in



the HP+LC and HP+HC conditions at LL intensity, the cells were, respectively,
transferred from 600 mL to 2350 mL bottles at the same time, and cultured for
another 2 days (Fig. S1b). Culture bottles were rotated 10 times until cells were mixed
at 09:00 h, 13:00 h and 19:00 h. Based on changes in cell densities during the
incubations, we calculated that at LL intensity, cells were acclimated to HP+LC and
HP+HC conditions for 10 generations. In the second day of the main experimental
cultures, subsamples were taken for measurements of cell densities, $pH_{Total}$, TA,
cellular contents of total particulate carbon (TPC), particulate organic carbon (POC),
nitrogen (PON) and phosphorus (POP), carbohydrate and protein. At the end of the
cultures under the previous conditions, cell samples with an initial density of 5000
cells ml$^{-1}$ were transferred from HP+LC condition (treatment 1) to LP+LC condition
(treatment 3), and from HP+HC condition (treatment 2) to LP+HC condition
(treatment 4) at LL intensity. The cells were acclimated to LP+LC and LP+HC
conditions for 8 generations before subsamples were taken for measurements.
At HL intensity (Part 2), samples grown under the HP+LC and HP+HC conditions
were transferred from 40 (LL) to 300 μmol photons m$^{-2}$ s$^{-1}$ (HL) of PAR with initial
cell density of 5000 cells ml$^{-1}$. The cells were cultured under the HP+LC and HP+HC
conditions for 2 days, respectively, and then diluted back to the initial cell density.
These processes were performed three times in 600 mL bottles at HL intensity, and
then the main experimental cultures were conducted in 2350 mL bottles. The cells
were, respectively, acclimated to HP+LC and HP+HC conditions for at least 8
generations at HL intensity. On the second day of the incubation, subsamples were
taken for measurements of the parameters. After that, cell samples with an initial
density of 5000 cells ml$^{-1}$ were transferred from HP+LC condition (treatment 5) to
LP+LC condition (treatment 7), and from HP+HC condition (treatment 6) to LP+HC





condition (treatment 8). At HL intensity, cell samples were acclimated for at least 8
generations in LP+LC and LP+HC conditions, respectively, before subsamples were
taken for measurements.

**2.2 Phosphate concentration and carbonate chemistry measurements**
In the beginning and on the second day of the incubations, samples for determinations
of phosphate concentration (20 mL), $pH_{Total}$ value (20 mL) and total alkalinity (TA)
(50 mL) were, respectively, filtered (PTFE filter, 0.22 μm pore size, Nantong) 7 h
after the onset of the light period (at 13:00). Dissolved inorganic phosphorus (DIP)
concentration was measured using a spectrophotometer (SP-722, Shanghai Spectrum
Instruments) following the phosphomolybdate method (Hansen and Koroleff, 1999).
The bottle for pH measurement was filled from bottom to top with overflow and
closed without a headspace. The $pH_{Total}$ value was measured immediately at 20°C
using a pH meter which was corrected with a standard buffer of defined pH in
seawater (Dickson, 1993). TA samples were treated with 10 μL saturated $HgCl_2$
solution and stored in the dark at 4.0°C, and TA was measured at 20°C by
potentiometric titration (AS-ALK1+, Apollo SciTech) according to Dickson et al.
(2007). Carbonate chemistry parameters were estimated from TA and $pH_{Total}$ using the
CO2SYS program of Pierrot et al. (2006) with carbonic acid constants, $K_1$ and $K_2$,
taken from Roy et al. (1993).

**2.3 Cell density and elemental content measurements**
Twenty milliliter samples to monitor the cell density were taken daily at 13:30 h, and
fresh media with the same DIP concentration and carbonate chemistry as in the initial
treatment conditions were added as top-up. Cell densities were determined using a



Multisizer$^{TM}$3 Coulter Counter (Beckman Coulter). Growth rates were calculated for
each replicate according to the equation: $\mu = (\ln N_t - \ln N_0) / d$, where $N_t$ and $N_0$ refer
to the cell densities on the second day and in the beginning of the main experiment,
respectively, and $d$ was the growth period in days.

After mixing, samples for determinations of TPC (300 mL), POC and PON (300

mL), and POP (300 mL) were obtained by filtering onto GF/F filters (precombusted at
450$^{o}$C for 6 h) at the same time (14:00 h) in each treatment. For POC and PON
measurements, samples were fumed with HCl for 12 h to remove inorganic carbon.
TPC, POC and PON samples were dried at 60$^{o}$C for 12 h and analyzed using an
Elementar CHNS analyzer (Vario EL cube, GmbH, Germany). Cellular particulate
inorganic carbon (PIC) content was calculated as the difference between cellular TPC
and POC contents (Fabry and Balch, 2010). To remove dissolved inorganic
phosphorus from the GF/F filters, POP samples were rinsed three times with 0.17 mol
L$^{-1}$ Na$_2$SO$_4$. After that, 2 mL 0.017 mol L$^{-1}$ MgSO$_4$ solution was added onto filters,
and POP samples were dried at 90$^{o}$C for 12 h, and then combusted at 500$^{o}$C for 6 h to
remove POC and digested by 0.2 mol L$^{-1}$ HCl (Solórzano and Sharp, 1980).
Phosphorus concentration was measured using a microplate reader (Thermo Fisher)
following the ammonium molybdate method (Chen et al., 1956) using
adenosine-5-triphosphate disodium trihydrate as a standard.

**2.4 Protein and carbohydrate measurements**
Samples for determinations of protein (600 mL) and carbohydrate (600 mL) were,
respectively, filtered onto polycarbonate filters (0.6 μm pore size, Nuclepore,
Whatman) and onto precombusted GF/F filters at 14:30 h. Protein samples were
extracted by bead milling (FastPrep Lysing Matrix D) in 0.5 mL 1× protein extraction



buffer (lithium dodecyl sulfate, ethylene diamine tetraacetic acid, Tris, glycerol and
4-(2-aminoethyl) benzenesulfonyl fluoride hydrochloride). Bead milling was
performed four times for 1 min at $6.5\ \mathrm{ms^{-1}}$, and samples were placed on ice for 2 min
between each round of bead milling to prevent degradation. The samples were then
centrifuged at $10,000 \times g$ for 5 min (Centrifuge 5418 R, Eppendorf, Germany), and
extracted protein in the supernatant was quantified using the BCA assay with bovine
gamma globulin as a standard using a microplate reader (Ni et al., 2016).
Carbohydrate samples were hydrolyzed with $12.00\ \mathrm{mol\ L^{-1}}$ $H_2SO_4$ in the dark for 1 h
and diluted by Milli-Q water to a final $H_2SO_4$ concentration of $1.20\ \mathrm{mol\ L^{-1}}$. Then,
samples were sonicated for 5 min, vortexed for 30 s, and boiled at 90°C for 3 h
(Pakulski and Benner, 1992). The extracted carbohydrate was determined by phenol
sulfuric reaction with D–glucose as standard (Masuko et al., 2005).

**2.5 Data analysis**
The percentages of carbon and nitrogen contributed by carbohydrate and protein were
calculated from the elemental composition of biochemical classes compiled by Geider
and LaRoche (2002). A three-way ANOVA was used to determine the main effects of
DIP concentration, light intensity and $CO_2$ level, and their interactions on each
variable. A Tukey post hoc test was performed to identify significant differences
between two DIP concentrations, two light intensities and two $CO_2$ levels. A
Shapiro–Wilk test was conducted to analyze the normality of residuals and a Levene
test was conducted graphically to test for homogeneity of variances. All data analyses
were conducted using the statistical software *R* with the packages carData, lattice and
nlme (R version 3.5.0).

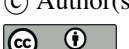



**3 Results**

**3.1 Dissolved inorganic phosphorus concentration and carbonate chemistry parameters**

During the incubations, organismal activity significantly reduces dissolved inorganic phosphorus (DIP) concentrations (Table 1). Under high phosphorus (HP) treatment, DIP concentrations decrease by 20.32% in low light (LL) and low $CO_2$ (LC), by 22.32% in LL and high $CO_2$ (HC), by 27.66% in high light (HL) and LC, and by 31.58% in HL and HC. Under low phosphorus (LP) treatment, DIP concentrations decrease from 0.43 μmol $L^{-1}$ at the beginning of the experiment to be lower than 0.04 μmol $L^{-1}$ (the detection limit) at the end of the incubation in LL and LC conditions, in LL and HC conditions, in HL and LC conditions, and in HL and HC conditions.

During the incubations, at LL intensity, $pH_T$ values increase by, on average, 0.02 in HP+LC, by 0.03 in HP+HC, by 0.09 in LP+LC, and by 0.10 in LP+HC conditions (Table 1). At HL intensity, $pH_T$ values increase by 0.05 in HP+LC, by 0.06 in HP+HC, by 0.12 in LP+LC, and by 0.09 in LP+HC conditions. Correspondingly, at LL intensity, seawater $CO_2$ concentrations decrease by 5.53% in HP+LC, by 6.89% in HP+HC, by 22.76% in LP+LC, and by 22.77% in LP+HC. At HL intensity, seawater $CO_2$ concentrations decrease by 16.18% in HP+LC, by 16.41% in HP+HC, by 28.92% in LP+LC, and by 22.30% in LP+HC. Overall, organismal activity has larger effects on carbonate chemistry under the LP treatment than under the HP treatment.

**3.2 Growth rate**

The effect of increasing light intensity on growth rate is positive which can be seen by comparing growth rate in the HL regimes with their paired LL regimes (Fig. 1a,b; Table 2), though the extent of increase in growth rate depends on $CO_2$ levels and



phosphate availability. Compared to LL intensity, growth rates at HL intensity
increased by 48.48% in HP+LC, by 50.87% in HP+HC, by 60.86% in LP+LC, and by
60.80% in LP+HC (Tukey post hoc test, all values of $p < 0.01$). The effect of
increasing $CO_2$ levels on growth rate depends on light intensity and phosphate
availability (Fig. 1a,b). Compared to LC level, growth rates in HC level decreased by
3.08% in LL and HP condition ($p = 0.48$), by 16.13% in LL and LP condition ($p <$
0.01), by 1.50% in HL and HP condition ($p = 0.68$), and by 16.27% in HL and LP
condition ($p < 0.01$). The effect of reduced phosphorus availability on growth rate is
negative and the extent of reduction in growth rate depends on light intensity and $CO_2$
levels (Fig. 1a,b). Compared to HP availability, growth rates in LP availability
decreased by 8.46% in LL and LC condition ($p <0.01$), by 20.81% in LL and HC
condition ($p < 0.01$), by 0.76% in HL and LC condition ($p = 0.99$), and by 15.63% in
HL and HC condition ($p < 0.01$). These results show that high $CO_2$ levels and low
phosphorus availability acted synergistically to reduce growth rate of *E. huxleyi*, and
increasing light intensity could partly counteract this response.

**3.3 POC, PON, POP and PIC contents**
The effect of increasing light intensity on POC content is positive, which was
observed by comparing POC content in all the HL regimes with their paired LL
regimes (Fig. 1c,d). The extent of increase in POC content depends on $CO_2$ levels and
phosphate availability. Compared to LL intensity, POC contents at HL intensity
increased by 27.15% in HP+LC, by 26.51% in HP+HC, by 43.24% in LP+LC, and by
58.13% in LP+HC conditions (Tukey post hoc test, all values of $p < 0.01$). The effect
of increasing $CO_2$ levels on POC content is light and phosphate dependent and can be
seen by comparing POC content in the HC regimes with their paired LC regimes (Fig.




1c,d). At LL intensity, POC contents are not significantly different between HP+LC,
HP+HC, LP+LC and LP+HC conditions (all values of $p > 0.1$). At HL intensity,
compared to LC level, POC contents in HC level increased by 5.12% in HP condition
($p = 0.74$), and by 8.28% in LP condition ($p = 0.07$). The effect of phosphate
reduction on POC content is light and $CO_2$ dependent, which can be seen by
comparing POC content in the LP regimes with those in their paired HP regimes (Fig.
1c,d). At LL intensity, POC contents did not significantly differ between LP and HP
availability. At HL intensity, compared to HP availability, POC contents in LP
availability increased by 11.80% in LC condition ($p = 0.02$), and by 15.28% in HC
condition ($p < 0.01$). These results show that ocean acidification and reduced
phosphorus availability acted synergistically to increase POC contents in HL
condition but not in LL condition.

The effect of increasing light intensity on PON content depends on $CO_2$ levels and

phosphate availability (Fig. 1e,f). Compared to LL intensity, PON contents at HL
intensity increased by 12.03% in HP+LC condition ($p = 0.27$), by 19.54% in HP+HC
condition ($p < 0.01$), by 22.68% in LP+LC condition ($p < 0.01$), and by 30.90% in
LP+HC condition ($p < 0.01$). The effect of increasing $CO_2$ levels on PON content is
light and phosphate dependent, which can be seen by comparing POC content in the
HC regimes with their paired LC regimes (Fig. 1e,f). Compared to LC level, PON
contents in HC level did not change significantly in LL and HP condition, in LL and
LP condition, in HL and LP condition, and increased by 14.68% in HL and HP
condition ($p = 0.02$). The effect of phosphate reduction on PON content is $CO_2$ and
light dependent, which can be seen by comparing PON content in the LP regimes with
those in their paired HP regimes (Fig. 1e,f). Compared to HP availability, PON
contents in LP availability decreased by 16.59% in LL and LC condition ($p = 0.01$),



by 24.03% in LL and HC condition ($p < 0.01$), by 8.35% in HL and LC condition ($p =$
0.43), and by 17.32% in HL and HC condition ($p < 0.01$). These results show that
increasing light intensity compensated for the negative effect of phosphate reduction
on PON content.

The effect of increasing light intensity on POP content is positive and can be seen

by comparing POP content in the HL regimes with their paired LL regimes, though
the extent of increase in POP content depends on $CO_2$ levels and phosphate
availability (Fig. 1g,h). Compared to LL intensity, POP contents at HL intensity
increased by 35.79% in HP+LC, by 41.70% in HP+HC, by 57.22% in LP+LC, and by
56.44% in LP+HC conditions (Tukey post hoc test, all values of $p < 0.01$). Ocean
acidification did not change the POP contents significantly in LL and HP condition, in
LL and LP condition, in HL and HP condition, and in HL and LP condition (all values
of $p > 0.53$) (Fig. 1g,h). Reduced phosphorus availability significantly decreased the
POP contents, which can be seen by comparing POP content in the LP regimes with
their paired HP regimes, though the extent of reduction in POP content depends on
light intensity and $CO_2$ levels (Fig. 1g,h). Compared to HP availability, POP contents
in LP availability decreased by 52.96% in LL and LC condition, by 54.03% in LL and
HC condition, by 46.11% in HL and LC condition, and by 49.51% in HL and HC
condition (all values of $p < 0.01$). These results show that reduced phosphorus
availability had a larger effect on POP content than that of ocean acidification and
increasing light intensity.

The effect of increasing light intensity on PIC content is positive, which can be

seen by comparing PIC content in the HL regimes with their paired LL regimes,
though the extent of increase in PIC content depends on $CO_2$ levels and phosphorus
availability (Fig. 1i,j). Compared to LL intensity, PIC contents at HL intensity





increased by 77.87% in HP+LC, by 70.28% in HP+HC, by 98.31% in LP+LC, and by
90.68% in LP+HC conditions (Tukey post hoc test, all values of $p < 0.01$). The effect
of increasing $CO_2$ levels on PIC content is negative and can be seen by comparing
PIC content in the HC regimes with those in their paired LC regimes (Fig. 1i,j). The
extent of reduction in PIC content depends on light intensity and phosphorus
availability. Compared to LC level, PIC contents under ocean acidification decreased
by 31.43% in LL and HP condition ($p = 0.09$), by 16.00% in LL and LP condition ($p$
$= 0.67$), by 35.02% in HL and HP condition ($p < 0.01$), and by 21.12% in HL and LP
condition ($p < 0.01$). The effect of phosphate reduction on PIC content is positive
which can be seen by comparing PIC content in the LP regimes with their paired HP
regimes, though the extent of increase in PIC content depends on light intensity and
$CO_2$ levels (Fig. 1i,j). Compared to HP availability, PIC contents in LP availability
increased by 16.00% in LL and LC condition ($p = 0.83$), by 41.26% in LL and HC
condition ($p = 0.16$), by 29.98% in HL and LC condition ($p < 0.01$), and by 60.44% in
HL and HC condition ($p < 0.01$). These results show that high light intensity and low
phosphorus availability acted synergistically to increase PIC content, which
counteracts the negative effect of ocean acidification on PIC content.

**3.4 Carbohydrate and protein contents**
The effect of increasing light intensity on carbohydrate content is positive and can be
seen by comparing carbohydrate content in the HL regimes with their paired LL
regimes, though the extent of increase in carbohydrate depends on $CO_2$ levels and
phosphorus availability (Fig. 2a,b). Compared to LL intensity, cellular carbohydrate
contents at HL intensity increased by 148.81% in HP+LC condition, by 139.42% in
HP+HC condition, by 179.12% in LP+LC condition, and by 204.42% in LP+HC



condition (all values of $p < 0.01$). The effect of increasing $CO_2$ levels on carbohydrate
content is light and phosphate dependent which can be seen by comparing
carbohydrate content in the HC regimes with their paired LC regimes (Fig. 2a,b).
Compared to LC level, carbohydrate contents under ocean acidification increased by
26.55% in LL and HP condition ($p = 0.58$), by 8.91% in LL and LP condition ($p =$
0.99), by 21.32% in HL and HP condition ($p = 0.02$), and by 18.45% in HL and LP
condition ($p < 0.01$). The effect of phosphate reduction on carbohydrate content is
light and $CO_2$ dependent and can be seen by comparing carbohydrate content in the
LP regimes with their paired HP regimes (Fig. 2a,b). Compared to HP availability,
carbohydrate contents in LP availability did not change significantly in LL and LC
condition, in LL and HC condition (both $p > 0.65$) and increased by 40.13% in HL
and LC condition ($p < 0.01$), and by 36.00% in HL and HC condition ($p < 0.01$).
These results show that increasing light intensity dominantly increased carbohydrate
content, and ocean acidification and reduced phosphorus availability acted
synergistically to increase carbohydrate contents under high light intensity.
The effect of increasing light intensity on protein content is positive, which can be
seen by comparing protein content in the HL regimes with their paired LL regimes,
though the extent of increase in protein content depends on $CO_2$ level and phosphorus
availability (Fig. 2c,d). Compared to LL intensity, protein contents at HL intensity
increased by 24.76% in HP+LC condition, by 30.43% in HP+HC condition, by
68.09% in LP+LC condition, and by 65.39% in LP+HC condition (all values of $p <$
0.01). The effect of increasing $CO_2$ levels on protein content can be seen by
comparing protein content in the HC regimes with their paired LC regimes (Fig. 2c,d).
Compared to LC level, protein contents under ocean acidification did not change
significantly in LL and HP condition, in LL and LP condition, in HL and HP




condition, and in HL and LP condition (all values of $p > 0.09$). The effect of
phosphate reduction on protein content is light and $CO_2$ dependent, which can be seen
by comparing protein content in the LP regimes with their paired HP regimes (Fig.
2c,d). Compared to HP availability, protein content in LP availability decreased by
27.88% in LL and LC condition, by 28.80% in LL and HC condition (both $p < 0.01$)
and did not change significantly in HL and LC condition, and in HL and HC condition
(both $p > 0.11$). These results show that high light intensity counteracted the negative
effect of low phosphorus availability on protein content, and ocean acidification had
less effect on protein content.

**3.5 Percentage of POC allocated to carbohydrate (carbohydrate–C : POC) and**
**protein (protein–C : POC)**
Increasing light intensity increased the percentage of POC allocated to carbohydrate
(carbohydrate–C : POC), which can be seen by comparing carbohydrate–C : POC in
the HL regimes with their paired LL regimes, though the extent of increase in
carbohydrate–C : POC depends on $CO_2$ levels and phosphorus availability (Fig. 2e,f).
Compared to LL intensity, carbohydrate–C : POC at HL intensity increased by
95.60% in HP+LC condition, by 97.69% in HP+HC condition, by 95.05% in LP+LC
condition, and by 83.37% in LP+HC condition (all values of $p < 0.01$). The effect of
increasing $CO_2$ levels on carbohydrate–C : POC is light and phosphate dependent, and
can be seen by comparing carbohydrate–C : POC in the HC regimes with their paired
LC regimes (Fig. 2e,f). Compared to LC level, carbohydrate–C : POC under ocean
acidification increased by 20.12% in LL and HP condition, by 11.42% in LL and LP
condition, by 20.36% in HL and HP condition, and by 4.40% in HL and LP condition
(all values of $p > 0.08$). The effect of phosphate reduction on carbohydrate–C : POC





is light and $CO_2$ dependent, which can be seen by comparing carbohydrate–C : POC
in the LP regimes with those in their paired HP regimes (Fig. 2e,f). Compared to HP
availability, carbohydrate–C : POC in LP availability increased by 25.61% in LL and
LC condition ($p = 0.16$), by 17.37% in LL and HC condition ($p = 0.47$), by 25.81% in
HL and LC condition ($p < 0.01$), and by 8.11% in HL and HC condition ($p < 0.05$).
These results show that increasing light intensity, ocean acidification and reduced
phosphorus availability acted synergistically to increase the percentage of POC
allocated to carbohydrate.
Increasing light intensity did not significantly change the percentage of POC
allocated to protein (protein–C : POC) in HP+LC, HP+HC, LP+LC and LP+HC
conditions (Fig. 2g,h). Ocean acidification did not significantly affect the protein–C :
POC in LL and HP, in LL and LP, in HL and HP, and in HL and LP conditions. The
effect of phosphate reduction on protein–C : POC is light and $CO_2$ dependent, which
can be seen by comparing the protein–C : POC in the LP regimes with their paired HP
regimes (Fig. 2g,h). Compared to HP availability, protein–C : POC in LP availability
decreased by 27.39% in LL and LC condition ($p < 0.01$), by 23.05% in LL and HC
condition ($p < 0.01$), by 12.81% in HL and LC condition ($p = 0.09$), and by 21.77% in
HL and HC condition ($p < 0.01$). These results show that reduced phosphorus
availability dominantly reduced the protein–C : POC, and increasing light intensity
and ocean acidification had less effects on protein–C : POC.

**3.6 Elemental stoichiometry and protein content as a function of growth rate**
At LL and HL intensities, both POC : POP ratio and PON : POP ratio were linearly
and negatively correlated with growth rates (Fig. 3a,b). In LL and HL conditions,
POC : POP ratio decreased linearly with increasing growth rate ($R^2 = 0.71$, $F = 32.08$,





$p < 0.01$ in LL condition; $R^2 = 0.53$, $F = 14.63$, $p < 0.01$ in HL condition). Similarly,
in LL and HL conditions, PON : POP ratio decreased linearly with increasing growth
rate ($R^2 = 0.69$, $F = 29.23$, $p < 0.01$ in LL condition; $R^2 = 0.50$, $F = 13.31$, $p < 0.01$ in
HL condition). In all treatments, protein content increased linearly with increasing
growth rate ($R^2 = 0.76$, $F = 151.14$, $p < 0.01$) (Fig. 3c), and POC content increased
linearly with increasing carbohydrate content ($R^2 = 0.94$, $F = 435.10$, $p < 0.01$) (Fig.
3d).

**4 Discussion**
Coccolithophores make an important contribution to marine biological carbon pump
and their responses to global climate change could have significant consequences for
marine carbon cycling (Riebesell et al., 2017). The bloom-forming coccolithophore *E.*
*huxleyi*, dominating the assemblages in seawater under limited phosphorus condition,
is likely to be exposed to increasing light intensity and ocean acidification in the
future ocean (Kubryakova et al., 2021). In this study, we observed that increasing
light intensity compensates for the negative effects of low phosphorus availability on
cellular protein and nitrogen contents (Figs. 1 and 2). Reduced phosphorus
availability, increasing light intensity and ocean acidification act synergistically to
increase cellular contents of carbohydrate and POC, and the allocation of POC to
carbohydrate. These regulation mechanisms in *E. huxleyi* could provide vital
information for evaluating carbon cycle in marine ecosystems under global change.

Ribonucleic acid (RNA) is the main phosphorus-containing macromolecule within

the cell (Geider and La Roche, 2002). Therefore, the reduced phosphorus availability
dominantly reduces the RNA content (Fig. S5), which contributes to low POP
contents (McKew et al., 2015) (Fig. 1g,h). In eukaryotic cells, ribosomal RNA (rRNA)



constitutes about 80% of the total RNA and is mainly used to create ribosome
(Dyhrman, 2016). Thus, reduced RNA contents decrease the numbers of ribosome,
which has a potential to decrease protein synthesis (Dyhrman, 2016; Rokitta et al.,
2016). On the other hand, low light intensity reduces the nitrate uptake and
assimilation efficiency of *E. huxleyi* and other phytoplankton species (Perrin et al.,
2016; Lu et al., 2018), which exacerbates the negative effect of low phosphorus
availability on protein synthesis and PON contents (Figs. 1e and 2c). Besides that, low
light intensity significantly reduces the rates of RNA synthesis, carbohydrate
synthesis and cell division (Zhang et al., 2021), which adds to the negative effect of
low phosphorus availability on growth rate of *E. huxleyi* (Fig. 1a). Under high light
intensity and low $CO_2$ level, reduced phosphorus availability did not change growth
rate and protein content (Figs. 1b and 2d), which suggests that *E. huxleyi* might
compensate for low phosphate-induced decreases in ribosome content by increasing
protein synthesis efficiency under increasing light intensity (Reith and Cattolico,
1985). Under high light intensity and ocean acidification, reduced phosphorus
availability did not significantly change protein content while reduced growth rate,
which might indicate the lowered protein synthesis efficiency (McKew et al., 2015).

Several studies report that reduced phosphorus availability (0.4–0.5 µmol $L^{-1}$) did

not change growth rates significantly during the short-time (2 or 3 days) incubations
under low $CO_2$ level and high light intensity (Rokitta et al., 2016; Zhang et al., 2020;
Wang et al., 2022) (Fig. S6). The reasons could be that *E. huxleyi* cells developed
high affinity for phosphate and increased the uptake rate of phosphate (Wang et al.,
2022) and could replace the phospholipid membrane with non-phosphorus membrane
during the short-time incubation of phosphorus limitation (Shemi et al., 2016). Our
data showed that reduced phosphorus availability and ocean acidification acted



synergistically to reduce growth rate under both low and high light intensities (Fig.
1a,b). One of the reasons could be that low pH value under ocean acidification up
regulates the expressions of a series of genes involved in ribosome metabolism, such
as genes of large subunit ribosomal protein L3, L38E, L30E (*RP-L3*, *RP-L38E*,
*RP-L30E*), and small subunit ribosomal protein S3E, S5E, SAE (*PR-S3E*, *PR-S5E*,
*RP-SAE*) in *E. huxleyi* (Fig. S7). Under ocean acidification, up regulation of
expression of these genes has the potential to drive cells to allocate more phosphorus
to synthesize ribosome, and to reduce the allocation of phosphorus to DNA
replication (Rokitta et al., 2011), which exacerbates the limitation of reduced
phosphorus availability on the rate of cell division in *E. huxleyi* (Rouco et al., 2013).
Under phosphorus-replete condition, more phosphorus is reallocated to ribosome
metabolism under ocean acidification which could facilitate nitrogen assimilation (Fig.
2d). Overall, under high light intensity, ocean acidification is likely to facilitate *E.*
*huxleyi* cells to increase nitrogen content in phosphorus-replete condition and to
reduce growth rate in phosphorus-limitation.
In this study, we found that low light intensity dominantly limits carbon
assimilation of *E. huxleyi* and reduces the effects of phosphate availability and ocean
acidification on carbohydrate and POC contents (Figs. 1c and 2a). However, under
high light intensity, *E. huxleyi* had high carbohydrate and POC contents while low
growth rate under reduced phosphorus availability and ocean acidification (Figs. 1b,d
and 2b), which suggests that carbon assimilation rate did not change significantly
while cell division rate decreased (Matthiessen et al., 2012; Perrin et al., 2016).
Furthermore, carbohydrate is a carbon- and energy-storing macromolecule (Geider
and La Roche, 2002). Under high light intensity, reduced phosphorus availability and
ocean acidification, *E. huxleyi* cells could synthesize more carbohydrate to store



carbon and energy, which contributes to the large percentage of POC allocated to
carbohydrate (Fig. 2f).

The physiological reasons for reduced calcification rate under ocean acidification

could be due to high proton concentration-induced reduction in $HCO_3^-$ uptake rate
(Meyer and Riebesell, 2015; Kottmeier et al., 2016). The molecular mechanisms for
low PIC content under ocean acidification may be due to down-regulation of a series
of genes potentially involved in ion transport and pH regulation, such as genes of
calcium/proton    exchanger    (*CAX3*),    sodium/proton    exchanger    (*NhaA2*)    and
membrane-associated proton pump (*PATP*) (Mackinder et al., 2011; Lohbeck et al.,
2014). On the other hand, increasing light intensity up-regulates a series of genes
related to ion transport, such as gene of *CAX3*, gene of $Cl^-/HCO_3^-$ exchanger and
genes of various subunits of a vacuolar $H^+$–ATPase (*V–ATPase*) and so on (Rokitta et
al., 2011). Up-regulation of these genes in high light intensity has the potential to
facilitate cells to take up $HCO_3^-$ and $Ca^{2+}$, and to pump proton outside the cells, and
then leads to large PIC content of *E. huxleyi* (Kottmeier et al., 2016). Our data suggest
that increasing light intensity counteracts the negative effect of ocean acidification on
PIC content of *E. huxleyi* (Fig. 1i,j). These results are consistent with the findings of
Feng et al. (2020) who reported that combinations of increasing light intensity and
ocean acidification increase the expression of genes involved in calcium-binding
proteins (*CAM* and *GPA*), which has the potential to increase calcium influx into cells
and then compensate for the effect of reduced $HCO_3^-$ uptake rate on calcification. It is
also suggested that increasing light intensity could cause cells to remove $H^+$ faster
which neutralizes the effect of high proton concentration on calcification (Jin et al.,
2017). On the other hand, reduced phosphorus availability extends the G1 phase of
cell cycle where calcification occurs, which prolongs the calcification time and then



increases PIC content (Müller et al., 2008). In addition, reduced phosphorus
availability up-regulates expressions of genes of $Ca^{2+}$ uptake, proton removal and
carbonic anhydrase, and then increases coccolith production (Wang et al., 2022),
which contribute to a larger PIC content and counteract the negative effect of ocean
acidification on PIC contents (Borchard et al., 2011) (Fig. 1i,j). Furthermore, one of
the reasons for larger PIC contents under reduced phosphorus availability and
increasing light intensity conditions are likely due to larger and more numerous
coccoliths (Gibbs et al., 2013; Perrin et al., 2016). Overall, responses of calcification
of *E. huxleyi* to ocean climate change are complex than previously thought (Meyer
and Riebesell, 2015), and it is worth exploring the underlying mechanisms of
calcification under changing multiple environmental drivers (Mackinder et al., 2011;
Feng et al., 2020).
Cellular POP content of *E. huxleyi* generally decreased, and POC : POP ratio and
PON : POP ratio increased with reducing phosphorus availability (Leonardo and
Geider, 2005; McKew et al., 2015). The negative correlations between growth rate
and POC : POP ratio or PON : POP ratio under each light intensity are consistent with
the growth rate hypothesis (Fig. 3), which proposes that growth rate increases with
increasing RNA : protein ratio. Phosphorus in RNA accounts for a high percentage of
total POP, whereas nitrogen in protein is the main form of PON (Zhang et al., 2021),
and the growth rate hypothesis suggests that growth rate could increase with
decreasing POC : POP ratio or PON : POP ratio (Sterner and Elser, 2002). Our results
suggest that *E. huxleyi* could reallocate chemical elements and energy to synthesize
carbohydrate, protein and RNA efficiently, and then regulate its elemental
stoichiometry and growth rate to acclimate to reduced phosphorus availability, ocean
acidification and increasing light intensity (Moreno and Martiny, 2018). In the future



ocean, large carbohydrate and POC contents, POC : PON ratio, and POC : POP ratio
of coccolithophores indicate increases in carbon export to the deep ocean that may
affect the efficiency of the biological carbon pump and the marine biogeochemical
cycle of carbon.



*Data availability.* The data are available upon request to the corresponding author
(yongzhang@fjnu.edu.cn).

*Author contributions.* YZ (yongzhang@fjnu.edu.cn), ZL and KX contributed to the
design of the experiment. YZ (yongzhang@fjnu.edu.cn), YZ
(qsx20211022@student.fjnu.edu.cn), SM, HC and RH performed this experiment and
biochemical analyses. YZ (yongzhang@fjnu.edu.cn) wrote the first manuscript draft.
All authors contributed to the data analyses and editing of the paper.

*Competing interests.* The authors declare that they have no conflict of interest.

*Acknowledgments.* We would like to thank Professor Zoe V. Finkel for providing the
*Emiliania huxleyi* RCC1266, Dr. Vinitha Ebenezer for her helpful revision of the
manuscript and two reviewers for their helpful suggestions which have help us to
improve the manuscript. This work was supported by the National Natural Science
Foundation of China (41806129 [YZ], 32001180 [ZKL]).




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

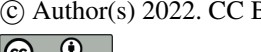



**Figure Legends**
**Figure 1.** Growth rate (**a**, **b**), cellular contents of particulate organic carbon (POC, **c**,
**d**), nitrogen (PON, **e**, **f**) and phosphorus (POP, **g**, **h**), and particulate inorganic carbon
(PIC, **i**, **j**) of *Emiliania huxleyi* RCC1266 in the treatments of high phosphorus
availability and low $CO_2$ level (HP+LC), high phosphorus availability and high $CO_2$
level (HP+HC), low phosphorus availability and low $CO_2$ level (LP+LC), and low
phosphorus availability and high $CO_2$ level (LP+HC) under low light (left, 40 μmol
photons $m^{-2}$ $s^{-1}$) and high light (right, 300 μmol photons $m^{-2}$ $s^{-1}$) intensities. Different
letters represent significant differences in each parameters between treatments ($p <$
0.05). The data represents the means and standard deviation of four independent
cultures.

**Figure 2.** Cellular contents of carbohydrate (**a**, **b**) and protein (**c**, **d**), and the
percentages of POC allocated to carbohydrate (**e**, **f**) and protein (**g**, **h**), and the
percentage of PON allocated to protein (**i**, **j**) of *E. huxleyi* RCC1266 in the treatments
of high phosphorus availability and low $CO_2$ level (HP+LC), high phosphorus
availability and high $CO_2$ level (HP+HC), low phosphorus availability and low $CO_2$
level (LP+LC), and low phosphorus availability and high $CO_2$ level (LP+HC) under
low light (left) and high light (right) intensities. Different letters represent significant
differences in each parameters between treatments ($p < 0.05$). The data represents the
means and standard deviation of four independent cultures. Please see figure 1 for
more detailed information.

**Figure 3.** Cellular POC : POP ratio (**a**), PON : POP ratio (**b**), and protein content (**c**)
of *E. huxleyi* RCC1266 as a function of growth rate, and cellular POC content as a





function of carbohydrate (**d**) in the treatments of high phosphorus availability and low
$CO_2$ level (HP+LC, □), high phosphorus availability and high $CO_2$ level (HP+HC,
○), low phosphorus availability and low $CO_2$ level (LP+LC, △), and low phosphorus
availability and high $CO_2$ level (LP+HC, ◇) under low light (LL, empty) and high
light (HL, fill) intensities. Each point indicates an individual replicate under each
treatment. Please see figure 1 for more detailed information.























**Table 1.** Carbonate chemistry parameters and dissolved inorganic phosphorus (DIP) concentration at the end of the incubation. The values are means ± standard deviation (sd) of four replicates. Respectively, LL and HL represent 40 and 300 μmol photons $m^{-2}$ $s^{-1}$ of photosynthetically active radiation (PAR), and HP and LP represent 4 and 0.43 μmol $L^{-1}$ $PO_4^{3-}$ at the beginning of the incubations.

| | | | $p$CO$_2$ (μatm) | pH (total scale) | TA (μmol L$^{-1}$) | DIC (μmol L$^{-1}$) | HCO$_3^-$ (μmol L$^{-1}$) | CO$_3^{2-}$ (μmol L$^{-1}$) | DIP (μmol L$^{-1}$) |
|----|----|----|----|----|----|----|----|----|----|
| LL | HP | LC | 403±4 | 8.06±0.01 | 2346±23 | 2074±21 | 1861±18 | 200±2 | 3.20±0.03 |
| | | HC | 881±20 | 7.77±0.01 | 2351±33 | 2216±32 | 2074±30 | 114±2 | 3.12±0.08 |
| | LP | LC | 329±4 | 8.13±0.01 | 2332±24 | 2024±22 | 1787±19 | 225±3 | <0.04 |
| | | HC | 730±8 | 7.84±0.01 | 2349±24 | 2189±23 | 2033±22 | 132±2 | <0.04 |
| HL | HP | LC | 357±9 | 8.09±0.01 | 2235±41 | 1959±35 | 1749±30 | 199±6 | 2.86±0.06 |
| | | HC | 791±7 | 7.80±0.01 | 2296±20 | 2151±20 | 2007±18 | 118±1 | 2.70±0.06 |
| | LP | LC | 303±4 | 8.16±0.01 | 2354±13 | 2024±11 | 1773±9 | 241±3 | <0.04 |
| | | HC | 735±19 | 7.83±0.01 | 2319±69 | 2162±65 | 2011±60 | 128±5 | <0.04 |

















**Table 2.** Growth rate ($d^{-1}$), cellular contents of POC, PON, POP, PIC, carbohydrate (CHO) and protein (Pro) (pg cell$^{-1}$), and the ratios of POC : PON, POC : POP, PON : POP and PIC : POC, and the percentages of POC allocated to carbohydrate (CHO–C : POC) and protein (Pro–C : POC), and the percentage of PON allocated to protein (Pro–N : PON) (%). LC and HC represent low $CO_2$ (426 µatm) and high $CO_2$ (946 µatm) levels, respectively. Please see table 1 for more detailed information.

| | Low light intensity | | | | High light intensity | | | |
| | HP | | LP | | HP | | LP | |
| | LC | HC | LC | HC | LC | HC | LC | HC |
|---|---|---|---|---|---|---|---|---|
| Growth rate | 0.91±0.03 | 0.88±0.01 | 0.83±0.02 | 0.70±0.03 | 1.35±0.03 | 1.33±0.02 | 1.34±0.02 | 1.12±0.02 |
| POC | 8.34±0.57 | 8.73±0.32 | 8.20±0.36 | 8.04±0.24 | 10.53±0.70 | 11.04±0.42 | 11.73±0.19 | 12.70±0.50 |
| PON | 1.49±0.16 | 1.58±0.12 | 1.23±0.06 | 1.20±0.10 | 1.65±0.05 | 1.89±0.07 | 1.51±0.07 | 1.56±0.06 |
| POP | 0.16±0.01 | 0.15±0.01 | 0.08±0.01 | 0.07±0.01 | 0.22±0.01 | 0.21±0.01 | 0.12±0.01 | 0.10±0.01 |
| PIC | 2.12±0.22 | 1.45±0.17 | 2.44±0.11 | 2.06±0.37 | 3.74±0.22 | 2.41±0.41 | 4.83±0.34 | 3.79±0.49 |
| POC:PON | 6.57±0.43 | 6.46±0.52 | 7.78±0.46 | 7.87±0.54 | 7.45±0.28 | 6.83±0.32 | 9.09±0.44 | 9.50±0.11 |
| POC:POP | 133.3±7.8 | 153.8±13.9 | 282.4±31.2 | 313.0±40.5 | 124.2±3.5 | 137.8±8.5 | 259.7±23.2 | 316.9±30.4 |
| PON:POP | 20.40±2.53 | 23.98±3.37 | 36.30±3.54 | 40.10±7.42 | 16.68±0.47 | 20.21±1.47 | 28.63±2.80 | 33.33±2.85 |
| PIC:POC | 0.26±0.04 | 0.17±0.02 | 0.30±0.02 | 0.26±0.04 | 0.36±0.02 | 0.22±0.04 | 0.41±0.03 | 0.30±0.04 |
| CHO | 1.45±0.15 | 1.81±0.16 | 1.79±0.16 | 1.94±0.16 | 3.58±0.41 | 4.30±0.17 | 4.96±0.24 | 5.85±0.49 |
| Protein | 5.23±0.55 | 5.37±0.39 | 3.73±0.27 | 3.80±0.15 | 6.45±0.36 | 6.97±0.22 | 6.25±0.29 | 6.28±0.30 |
| CHO–C:POC | 6.95±0.41 | 8.31±0.85 | 8.71±0.70 | 9.64±0.58 | 13.62±1.43 | 15.60±0.98 | 16.92±1.04 | 18.39±0.96 |
| Pro–C:POC | 33.26±3.24 | 32.58±1.98 | 24.15±2.52 | 25.07±1.69 | 32.49±1.69 | 33.51±1.41 | 28.23±1.35 | 26.21±1.27 |
| Pro–N:PON | 56.84±8.96 | 54.55±6.51 | 48.41±2.46 | 51.07±5.40 | 62.62±2.88 | 59.12±1.21 | 66.35±4.06 | 64.44±2.73 |






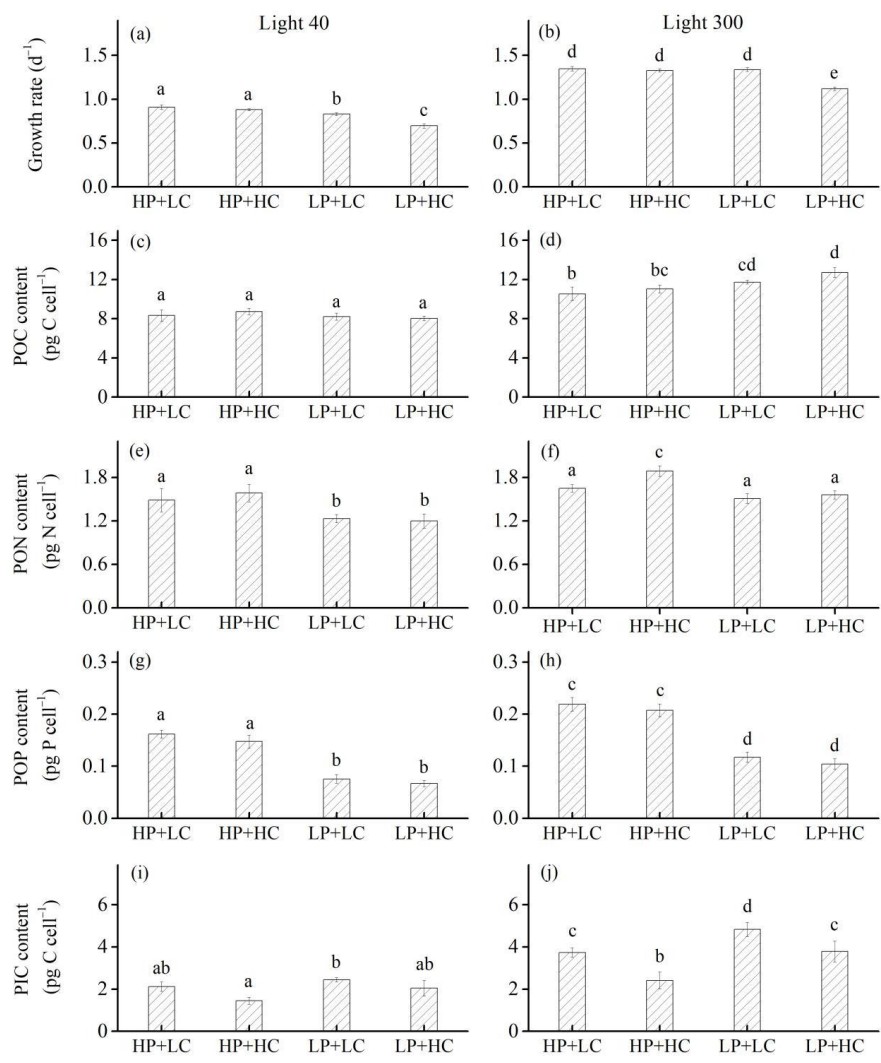



Figure 1







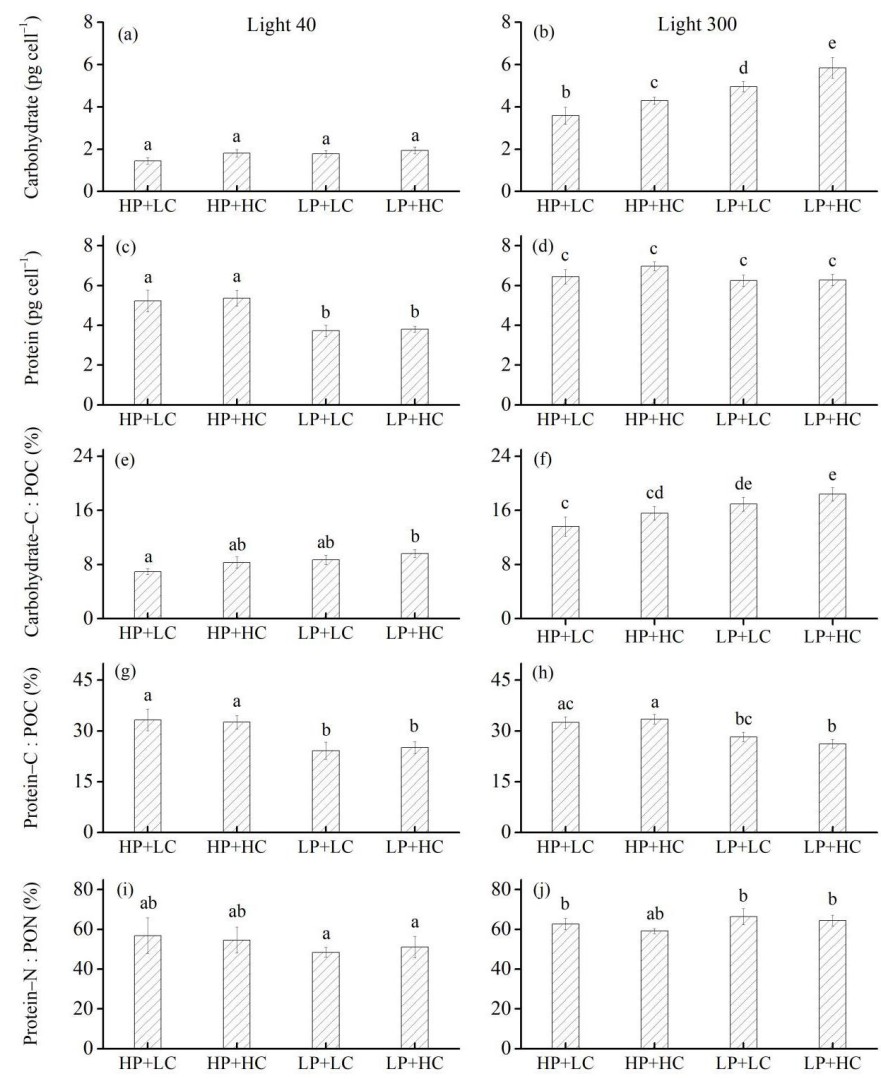



Figure 2





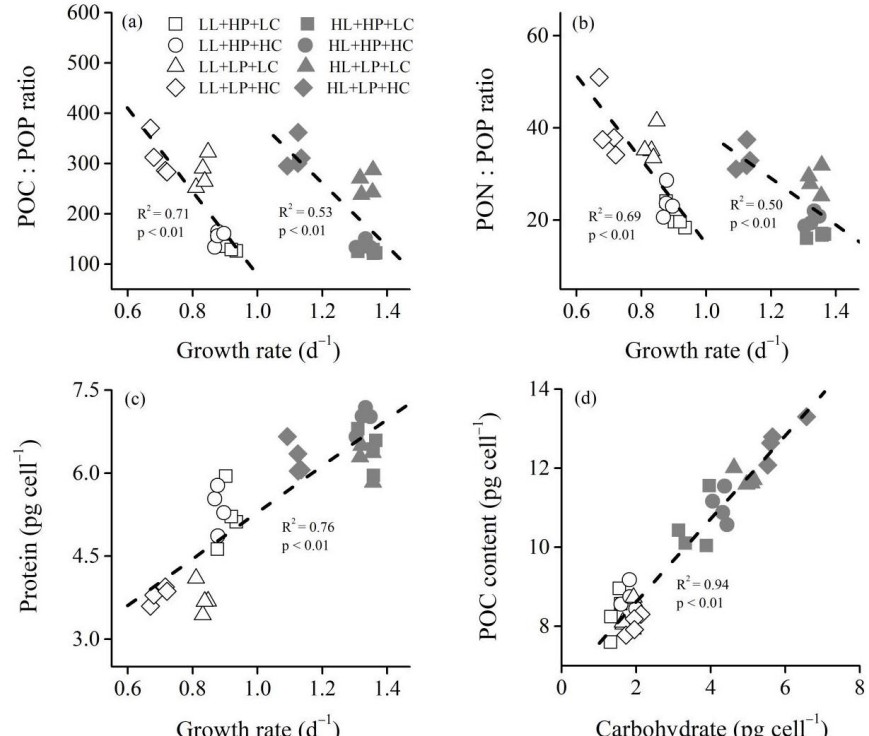



Figure 3
