# Peer review of "Reallocation of elemental content and macromolecule in the coccolithophore"

_EGUsphere, 2022_

## Author Response (AR1)

Dear Editor,

We thank the referees for their supportive and constructive comments to our manuscript. We have responded to the comments point by point as follows. The revised sentences or contents are underlined.

**Responses to comment 1 are following:**

**Comment 1**

The manuscript reported a practical finding that responses of elemental content and macromolecule of the *Emiliania huxleyi* to reduced phosphorus availability and ocean acidification depend on light intensity. The results of the study showed that under future ocean acidification and increasing light intensity conditions, enhanced carbon fixation could increase carbon storage in the phosphorus-limited regions of the oceans where *E. huxleyi* dominates the phytoplankton assemblages. This research could provide vital information for evaluating carbon cycle in marine ecosystems under global change.

The paper is well structured, the data are presented well, and the figures are well organized. However, some parts of the manuscript need further explanation and to be improved before acceptance for publication.

Response: We have revised the manuscript according to the suggestiongs from the referee.

Line 60-61, 63-64: Please specify the light levels for "low light" and "high light". Particulate organic nitrogen and protein contents at a cellular basis or on a total scale?

Response: We have revised this part as indicated at lines 60–66 in the marked-up manuscript version (below): "Reduced phosphorus availability reduces particulate organic nitrogen and protein contents per cell under 40 μmol photons m$^{-2}$ s$^{-1}$, but not under 300 μmol photons m$^{-2}$ s$^{-1}$. Reduced phosphorus availability and elevated $p$CO$_2$ concentrations act synergistically to increase particulate organic carbon (POC) and carbohydrate contents per cell under 300 μmol photons m$^{-2}$ s$^{-1}$ but not under 40 μmol

photons m$^{-2}$ s$^{-1}$".

Line 62: I was wondering if it is more appropriate to use the term of "elevated pCO2" rather than "ocean acidification", which has not been introduced in the text before. In fact, the authors used the terms of elevated pCO2 and ocean acidification (used in a broad sense to include effects of both elevated dissolved CO2 and the resulting decrease in pH) in the text, which is confusing. So, I would suggest the authors to select one term and use it consistently throughout the whole manuscript.

Response: Agree. We have checked through the whole manuscript, and found that the "elevated $p$CO$_2$" and "ocean acidification" have the same meaning. Thus, we do not change "ocean acidification" to "elevated $p$CO$_2$ concentrations" since the introduction of the manuscript. However, we have changed "ocean acidification" to "elevated $p$CO$_2$ concentrations" throughout the abstract.

Line 140-141: Under reduced phosphorus availability, increasing light intensity and ocean acidification conditions. Should be cellular POC content? Please check this issue throughout the whole manuscript.

Response: Agree. We have changed "POC content" to "cellular POC content", and changed "PIC content" to "cellular PIC content" throughout the whole manuscript.

Line 143: Cellular PIC content?

Response: We have changed "PIC content" to "cellular PIC content" throughout the whole manuscript.

Line 283-287 The method described in the paper is very similar to the phenol-sulfuric acid method for the determination of polysaccharides. Is the carbohydrate assay the same as the polysaccharide assay?

Response: Yes. In this study, polysaccharide was hydrolyzed to monosaccharide (glucose). Then we measured the total contents of glucose by using phenol-sulfuric reaction which can be used to measure the contents of polysaccharide.

Line158-214 The description of At LL intensity (Part 1) is too tedious and complicated, please simplify. Just explain the reasons for selecting the light intensity, phosphate concentration, acidification conditions and the approximate experimental

steps. Or put the picture of the experimental method in the supplement to the manuscript, after all, there are only three figures in the manuscript.

Response: Agree. In the marked-up manuscript version, we have deleted these contents: ", and the ratio was about 6.5 mL $CO_2$-saturated seawater : 1000 mL ASW media. The $CO_2$-saturated seawater was achieved by bubbling pure $CO_2$ gas into 500 ml ASW media with a total alkalinity of 2350 μmol $L^{-1}$ for 2 h." **at lines 191–194**, and "The volumes of culture inoculum were calculated to match the volumes of media taken out from the bottles prior to inoculation." **at lines 201–203**, and "in the HP+LC and HP+HC conditions at LL intensity," and "respectively" **at lines 208–209**, and "Based on changes in cell densities during the incubations, we calculated that at LL intensity, cells were acclimated to HP+LC and HP+HC conditions for 10 generations." **at lines 212–214**, and "The cells were, respectively, acclimated to HP+LC and HP+HC conditions for at least 8 generations at HL intensity." **at lines 230–232**.

Line 290-292 The percentages of carbon and nitrogen contributed by carbohydrate and protein were an important element of this study. Please provide more details of this method.

Response: Agree. We have added these contents "(C : carbohydrate is 40% and C : protein is 53%)" and "(N : protein is 16%)" at lines 300–301.

Fig 2 and 3, if the purpose of the authors is to compare the different responses between the light level 40 and 300 μmol photons $m^{-2}$ $s^{-1}$, it would be better to merge Fig 1a and 1b.

Response: Agree. The results under two light intensities are shown in a graph, and two columns are shown for the treatment of HP+LC, HP+HC, LP+LC, and LP+HC in Figures 1 and 2 at lines 1009–1018.

Line 330-333: The p-values (0.48 and 0.68) indicated that the differences are not statistically significant. Also see this issue at Lines 365, 407, 413-414, 430 etc.

Response: Agree. We added these contents "The significant difference between treatments was set at $p < 0.05$." at lines 308–309. We think that although the differences between treatments are not statistically significant, the response trends of each parameters are also important.

We have changed "by 0.76% in HL and LC condition ($p = 0.99$)," to ", and did

not change significantly in HL and LC condition ($p = 0.99$)" **at lines 348–350**, and changed "by 26.55% in LL and HP condition ($p = 0.58$), by 8.91% in LL and LP condition ($p = 0.99$)," to "did not change significantly under LL condition and" **at lines 444–446**.

Line 337: See my comment above.

Response: We have deleted these contents "by 0.76% in HL and LC condition ($p = 0.99$)," and added these contents ", and did not change significantly in HL and LC condition ($p = 0.99$)" at lines 348–350.

Line 338-340: As the author concluded that there was significant interaction between CO2 and P (acted synergistically), it would be better to present the results of three-way ANOVA analysis in the text to support this statement. This issue occurred at Lines 360-362, and Lines 415-417 (included but not limited, please check throughout the whole manuscript).

Response: Thanks. Table S1 showed the statistical results of the three-way ANOVA analysis for each parameters. So we have added "(Table S1)" throughout the result section (at lines 352, 375, 392, 410, 432, 456, 475, 499 and 512).

Line 385-388: Please rephrase the wordings here.

Response: Agree. We have changed "Ocean acidification did not change the POP contents significantly in LL and HP condition, in LL and LP condition, in HL and HP condition, and in HL and LP condition (all values of $p > 0.53$)." to "Ocean acidification did not change the POP contents significantly under all treatments used here (all values of $p > 0.53$)." at lines 399–401.

Line 484-487: Please rephrase the wordings here.

Response: Agree. We have changed "Increasing light intensity did not significantly change the percentage of POC allocated to protein (protein–C : POC) in HP+LC, HP+HC, LP+LC and LP+HC conditions" to "Increasing light intensity did not significantly change the percentage of POC allocated to protein (protein–C : POC) under the phosphorus availability and $CO_2$ levels used here" at lines 500–502.

Line 520-521: I cannot see any "regulation mechanisms" that were proposed in the

present study.

Response: We have changed "regulation mechanisms" to "changes" at line 543.

Q8. Line 524 'Therefore, the reduced phosphorus availability dominantly reduces the RNA content (Fig. S5)'. Was the algal cell RNA content also measured in this study? The method for this determination was not described in the M&M section, at least.

Response: Yes, we also measured total cellular RNA content in this study. Total RNA was extracted by using TRIzol reagent (Invitrogen) and the RNeasy Plus Mini Kit (Qiagen). Samples were resuspended in 1.2 mL TRIzol reagent (Thermo Fisher Scientific) in lysing matrix D tubes (MP Biomedicals), homogenized by a FastPrep-24 machine (MP Biomedicals, 3 cycles, 8.0 m s$^{-1}$, 30 s, 3 min ice-chilling at interval), followed by centrifugation at 12,000 × g for 3 min at 4 °C (Eppendorf 5430R). RNA was extracted using a standard phenol-chloroform method (Chomczynski and Sacchi 1987). For more detail information, please see Zhang et al. (2021). These contents were added in the supplemental information.

Chomczynski, P., and Sacchi, N.: Single-step method of RNA isolation by acid guanidinium thiocyanate-phenol-chloroform extraction, Anal. Biochem., 162, 156–159, doi: 10.1006/abio.1987.9999, 1987.

Zhang, Y., Li, Z. K., Schulz, K. G., Hu, Y., Irwin, A. J., and Finkel, Z. V.: Growth-dependent changes in elemental stoichiometry and macromolecular allocation in the coccolithophore *Emiliania huxleyi* under different environmental conditions, Limnol. Oceanogr., 66, 2999–3009, doi: 10.1002/lno.11854, 2021.

Line 545: '(Rokitta et al., 2016; Zhang et al., 2020; Wang et al., 2022) (Fig. S6)'?
Response: We have deleted "Fig. S6" at line 574.

Line 555 '…..and small subunit ribosomal protein S3E, S5E, SAE (PR-S3E, PR-S5E, RP-SAE) in E. huxleyi (Fig. S7)'. Please provide references for this statement.
Response: Agree. Wilson and Doudna Cate (2012) reported the structure and function of the eukaryotic ribosome. So we have cited "(Wilson and Doudna Cate, 2012)" at line 584.

Line 574: The authors found that more POC were allocated to carbohydrates but not to protein under high light intensity, reduced phosphorus availability and ocean acidification. This is a quite interesting finding. So, what is the possible reason for this? Could you please explain?

Response: Carbohydrate is a carbon- and energy-storing macromolecule, and protein is related to growth rate. Under high light intensity, reduced phosphorus availability and ocean acidification, growth rate did not change, so cells did not necessary to increase cellular protein content whileas they can synthesize more carbohydrate to store carbon and energy. We have added these contents ", and protein is related to growth rate" at lines 601–602, and added "but didn't increase protein content" at lines 604–605.

**Responses to comment 2 are following:**

**Comment 2**

Dr Zhang and colleagues evaluate the interactive effect of changes on multiple environmental drivers (namely phosphorous concentration, pCO2 and light intensity) on the elemental and macromolecule content of the model coccolithophore species *Emiliania huxleyi*. Authors identify that multiple stressors interact on the physiological response of *E. huxleyi* and that underlying mechanisms are complex. For example, authors identify that reduced phosphorus availability results in a reduction of both particulate organic nitrogen and protein contents under low light intensity, but not under high light intensity. Interestingly, reduced phosphorus concentration in combination with ocean acidification result in an increase in both POC and carbohydrate contents only under high light conditions. A large body of evidence indicate that coccolithophores are sensitive to projected changes in oceanic conditions driven by ongoing human-induced climate change, such as ocean acidification and changes in nutrient supply, mixed layer depth and light intensity. Given their abundance and fundamental role in the biological and carbonate counter pumps, changes in coccolithophore performance will most likely have impacts in the oceanic carbon cycle and marine ecosystems. Therefore, there is an urgent need of studies such as the one presented here to evaluate how multiple environmental drivers affect key marine organism in order to predict how ongoing environmental change will impact marine ecosystems. The paper is clearly written, the figures are appropriate and the findings interesting and useful for the scientific community. Therefore, I recommend acceptance of this manuscript after the comments listed below have been addressed.

Response: We thank the referee for his/her comments.

Lines 84-85. The sentence "These ocean changes expose phytoplankton cells within the UML to multiple drivers," is vague. Do authors mean that environmental changes in the UML will expose phytoplankton cells to physiological stress? Please clarify.

Response: Thanks. We have changed "These ocean changes expose phytoplankton cells within the UML to multiple drivers" to "Environmental changes in the UML will expose phytoplankton cells to physiological stress" at lines 86–88.

Line 109. Authors could dedicate a sentence or two to the importance/relevance of multi-stressor experiments highlighting that numerous environmental factors will simultaneously change in the future ocean and therefore, this kind of experiments are needed to evaluate the response of different organisms to ongoing environmental change.

Response: Agree. We have added "In the future ocean, numerous environmental factors will simultaneously change and the extent of these changes may increase (Gao et al., 2019)." at lines 109–110.

Line 149. The specialized reader would appreciate an SEM picture of the coccosphere and or coccoliths either in the main text or as supplementary material.

Response: Agree. We have presented an SEM picture of the coccosphere and coccolith in the supplemental information.

[Figure]

**Figure S1.** A scanning electron microscopy (SEM) picture of the coccosphere and coccolith of *Emiliania huxleyi* RCC 1266.

Line 151. could authors mention the date when this strain was isolated?

Response: *Emiliania huxleyi* RCC1266 was isolated in 2007. We have added "in 2007" at line 156. For more detail information, please link to the website: https://roscoff-culture-collection.org/rcc-strain-details/1266

Line 510 Coccolithophores play a complex role in the carbon cycle through production and export of organic carbon to depth but also through the carbonate counter pump (releasing CO2 during the calcification process). Authors should clarify this in the text.

Response: Agree. We have changed "make an important contribution to marine biological carbon pump" to "play a complex role in the marine carbon cycle through production and export of organic carbon to depth but also through the carbonate counter pump" at lines 531–533.

Line 530. Could authors explain (or speculate) about the metabolic process behind the relationship between low light intensity and nitrate uptake reduction?

Response: In fact, our data showed that compared to saturation light intensity, the expression of genes related to nitrate reductase and nitrite reductase down-regulated significantly at limiting low light intensity. This is one of the reasons for low rate of nitrate uptake under low light intensity. In addition, Lu et al. (2018) reported that decreasing light intensity reduced the $N_2$ fixation rate of *Trichodesmium*. So we have added "Low light intensity down-regulates the expression of genes related to nitrate reductase and nitrite reductase, and then" at lines 555–556.

**Responses to comment 3 are following:**

**Comment 3**

December 21, 2022

Reviewer's comments for

Ms. Ref No.: egusphere-2022-947

Title: Responses of elemental content and macromolecule of the coccolithophore Emiliania huxleyi to reduced phosphorus availability and ocean acidification depend on light intensity

Authors: Yong Zhang et al.

Submitted to: Biogeosciences

General comments

The authors show interesting experimental data on the responses of an important coccolithophore to changing three environmental drivers. Culture experiments are carefully designed and conducted. Results are complex and result descriptions can be more accomplished but acceptable. I found some concern in the present manuscript. The following comments should be considered before this paper being considered for publishing in Biogeosciences.

Response: We thank the referee for his/her comments.

Major comments

1. I consider that a message from the present title (the importance of light condition on the coccolithophore responses) does not link with the conclusion of this paper (reallocation of material and energy to acclimate to climate change).

Response: Agree. We have changed the title to "Reallocation of elemental content and macromolecule in the coccolithophore *Emiliania huxleyi* to acclimate to climate change" at lines 1–3.

2. Results section. In cases of the difference between treatments being not statistically significant, one case is described as increase or decrease (e.g. line 331), another case is described as not significantly different (e.g. line 351). What is the threshold for authors to change the descriptions? I consider that this may lead readers

to the direction of authors' thought.

Furthermore, the authors show statistical information on interactions in three-way ANOVAs in Table S1, but these information are lacking in the text.

Response: Thanks. In this study, we described our results as increase or decrease (namely the response trends in each parameters under climate change condition). When the measured parameters were not significantly different under all treatments, the results were described as not significantly different.

We have added "The significant difference between treatments was set at $p <$ 0.05." at lines 308–309. Table S1 showed the statistical results of the three-way ANOVA analysis for each parameters. So we have added "(Table S1)" throughout the result section (at lines 352, 375, 392, 410, 432, 456, 475, 499 and 512).

3. Figures 3a and 3b, and L615-617. If the growth rate hypothesis is working here, all data lay on a line as shown in Figure 3C. Different P storage contents, as shown below, between LL and HL may cause the separation of the regression line among light conditions. Similar discussion may be required to interpret the results for POP contents (not organic actually I think).

Response: Thanks. We have added "On the other hand, the separation of the regression line between POC : POP ratio (or PON : POP ratio) and growth rate under low and high light intensities suggests different POP storage contents in *E. huxleyi* among different light intensities (Perrin et al., 2016). " at lines 552–555.

Specific comments

1. To confirm carbonate chemistry in some case, salinity data are needed.

Response: Agree. We have added "with a salinity of 33 psu, " at lines 161.

2. Is the low DIP 0.43 umol/L an upper end of DIP conc in coastal waters? I do not consider this as a general case in coastal waters.

Response: Agree. We have changed "and low DIP concentration corresponds to the upper end of the range of phosphate concentration in the coastal waters (Larsen et al., 2004)" to "at the end of the incubation, low DIP concentration limits the growth of *E. huxleyi* (see below)." at lines 178–181.

3.  Authors' selection on total boron formulation also may be important.

Response: Agree. We have added "a boron concentration of 372 μmol L$^{-1}$" at line 161.

4.  I understand the procedures remove PIC, so the measurements represent POC. On the other hand, for nitrogen and phosphorus inorganic forms are not removed from the filter samples, so these are not PON and POP but total particulate N and P. This may be critical to understand the variations in cellular contents of N and P. In particular, P storages occur in inorganic forms, and environmental drivers can alter the P storage capacity of the cells.

Response: Agree. Environmental drivers may alter the P storage capacity of *E. huxleyi* cells. In this study, one sample on the GF/F filter was fumed with HCl, dried and then used to measure the POC and PON contents. So the PON content is the particulate organic nitrogen. In addition, POP samples are rinsed three times with 0.17 mol L$^{-1}$ Na$_2$SO$_4$ to remove dissolved inorganic phosphorus from the GF/F filters. So we think that the POP content is also the particulate organic phosphorus.

5.  I found no descriptions on protein-N:PON ratios.

Response: We have added "On the other hand, increasing light intensity, reduced phosphorus availability and ocean acidification did not significantly change the percentage of PON allocated to protein (protein–N : PON) (Fig. 2e)." at lines 512–515.

6.  Results for POC:PON ratios are shown in Table S1 and Figure S4, but why are these not described in the text?

Response: Thanks. We have added "Reduced phosphorus availability increased the POC : PON ratio, and the extent of the increase was larger under HL than LL intensity (Fig. S5a)." at lines 518–519.

7.  The main cultures are conducted in 2 days, but authors conducted three pre-experimental cultures, resulting at least 8 generations acclimations. Therefore cells have enough time periods to change growth rate against the low phosphate conditions.

Response: Agree. We have added ", which allows cells to have enough time periods to change growth rate against the low DIP concentration." at lines 222–224.

8.  POC:PON is not discussed in Results and Discussion sections, so this discussion is sudden and not readily acceptable for readers.

Response: Thanks. We have added "Reduced phosphorus availability increased the POC : PON ratio, and the extent of the increase was larger under HL than LL intensity (Fig. S5a)." at lines 518–519.

9.  L625-629. This discussion on carbon cycle is not match with this paragraph. A new paragraph should be made for this discussion.

Response: We agree with the suggestions from the referee. A new paragraph is made to show the effect of coccolithophores on the marine carbon cycle at lines 655–662: "In the future ocean, large carbohydrate and POC contents, POC : PON ratio, and POC : POP ratio of coccolithophores indicate increases in carbon export to the deep ocean that may affect the efficiency of the biological carbon pump and the marine biogeochemical cycle of carbon (Meyer and Riebesell, 2015). In addition, increased cellular PIC content under phosphorus limitation condition may have the potential to weaken $CO_2$ uptake of the oceans in phosphorus-limited marine environments. In summary, responses of coccolithophores to climate change is likely to affect the marine carbon cycle in the future (Riebesell et al., 2017)."

10.  Figures and Tables. For this kind of experiments, data should be shown as a mean and standard error.

Response: Agree. In this study, all data are shown as the means and standard deviation of four independent cultures.

11.  Figures 1 and 2. I understand that statistical analyses are performed including two different light conditions (e.g. Figure 1a and b). I consider that two light conditions should be shown in a graph. Two columns can be shown for each treatment (e.g. HP+LC).

Response: Agree. The results under two light intensities are shown in a graph, and two columns are shown for the treatment of HP+LC, HP+HC, LP+LC, and LP+HC in Figures 1 and 2 at lines 1009–1018.

12. Figure S4. The label of treatments for right edge in each panel should be "LP+HC" but not "LP+LC".

Response: Thanks. We have changed "LP+LC" to "LP+HC" in Figure S5.

13. Figure S6. I do not understand the meanings for the authors showing these graphs.

Response: Agree. We have deleted the Figure S6 in the old version of the supplemental information.

14. Figure S7. No descriptions on methodology for the gene analyses are found in the text.

Response: Agree. The quality of the raw reads was assessed by Fastqc v.0.11.8 (Andrews, 2010) and Fastq screen v.0.13.0 (Wingett and Andrews, 2018) and summarized using Multiqc (Ewels et al., 2016). Trimming of the raw reads was performed to remove low-quality bases and adapter sequences with Trimmomatic v.0.38 (Bolger et al., 2014). *De novo* transcriptome assembly was performed with the Trinity's version 2.11.0 (Grabherr et al., 2011), and the low-quality assembly were removed with CD-HIT (Li and Godzik, 2006). A preliminary assessment of *de novo* assembly quality was performed with Transrate (Smith-Unna et al., 2016) and Busco (Hara et al., 2015), and the completeness assessment yielded high scores for all assemblies. Open reading frames (ORFs) were then predicted using TransDecoder version v5.5.0 (Haas et al., 2013), and were then annotated by Blastx, Hmmpress, Signalp, Rnammer, PFam (Lagesen et al., 2007). All annotations were loaded and integrated with Trinotate v3.0.0 (Haas et al., 2013). ORFs were further functionally annotated and assigned to the KEGG and GhostKOALA (Moreno-Santillán et al., 2019). Cleaned and trimmed reads of each sample were mapped to the assembled transcriptomes by salmon v0.9.1 (Patro et al., 2017). The differential expression was them calculated by DESeq2 v1.24.0 (Love et al., 2014) with a Benjamini-Hochberg adjusted $p$-value < 0.05. Data analysis and visualizations were made using R v.3.6.1 (Team, 2020), packages ggplot2 v.3.2.0 (Wickham, 2016) and Pheatmap v.1.0.12 (Kolde, 2015). We have added these contents in the supplemental information.

References

Andrews, S.: FASTQC: a quality control tool for high throughput sequence data, Cambridge, 2010. https://www.bioinformatics.babraham.ac.uk/projects/fastqc/.

Bolger, A. M., Lohse, M., and Usadel, B.: TRIMMOMATIC : a flexible trimmer for illumina sequence data, Bioinformatics, 30, 2114–2120, 2014.

Chomczynski, P., and Sacchi, N.: Single-step method of RNA isolation by acid guanidinium thiocyanate-phenol-chloroform extraction, Anal. Biochem., 162, 156–159, doi: 10.1006/abio.1987.9999, 1987.

Ewels, P., Magnusson, M., Lundin, S., and Käller, M.: MULTIQC: summarize analysis results for multiple tools and samples in a single report, Bioinformatics, 32, 3047–3048, 2016.

Grabherr, M. G., Haas, B. J., Yassour, M., Levin, J. Z., Thompson, D. A., Amit, I., Adiconis, X., Fan, L., Raychowdhury, R., Zeng, Q., Chen, Z., Mauceli, e., Hacohen, N., Gnirke, A., Rhind, N., di Palma, F., Birren, B. W., Nusbaum, C., Lindblad-Toh, K., Friedman, N, and Regev, A.: Full-length transcriptome assembly from RNA-Seq data without a reference genome, Nat. Biotechnol., 29, 644–652, 2011.

Haas, B. J., Papanicolaou, A., Yassour, M., Grabherr, M., Blood, P. D., Bowden, J., Couger, M. B., Eccles, D., Li, B., Lieber M., MacManes, M. D., Ott, M., Orvis, J., Pochet, N., Strozzi, F., Weeks, N., Westerman, R., Willian, T., Dewey, C. N., Henschet, R., LeDuc, R. D., Friedman, N., and Regev, A: *De novo* transcript sequence reconstruction from RNA-seq using the Trinity platform for reference generation and analysis, Nat. Protoc., 8, 1494–1512, 2013.

Hara, Y., Tatsumi, K., Yoshida, M., Kajikawa, E., Kiyonari, H., and Kuraku, S.: Optimizing and benchmarking *de novo* transcriptome sequencing: from library preparation to assembly evaluation, BMC Genomics, 16, 977, 2015.

Kolde, R., and Kolde, M. R.: Package 'pheatmap'. R. Package, 1, 790, 2015.

Lagesen, K., Hallin, P., Rodland, E. A., Staerfeldt, H. H., Rognes, T., and Ussery, D. W.: RNAmmer: Consistent and rapid annotation of ribosomal RNA genes, Nucleic Acids Res., 35, 3100–3108, 2007.

Li, W., and Godzik, A.: Cd-hit: A fast program for clustering and comparing large sets of protein or nucleotide sequences, Bioinformatics, 22, 1658–1659, 2006.

Love, M. I., Huber, W., and Anders, S.: Moderated estimation of fold change and dispersion for RNA-seq data with DES EQ 2, Genome Biol., 15, 550, 2014.

Moreno-Santillán, D. D., Machain-Williams, C., Hernández-Montes, G., and Ortega, J.: *De novo* transcriptome assembly and functional annotation in five species of bats, Sci. Rep., 9, 6222, doi: 10.1038/s41598-019-42560-9, 2019.

Patro, R., Duggal, G., Love, M. I., Irizarry, R. A., and Kingsford, C.: Salmon provides fast and bias-aware quantification of transcript expression, Nat. Methods., 14, 417–419, 2017.

R Development Core Team: R: a language and environment for statistical computing, v. 4.0.3. [WWW document] URL www.r-project.org [accessed 10 October 2020].

Smith-Unna, R., Boursnell, C., Patro, R., Hibberd, J. M. and Kelly, S.: TransRate: Reference-free quality assessment of *de novo* transcriptome assemblies, Genome Res., 26, 1134–1144, 2016.

Wickham, H.: ggplot2: elegant graphics for data analysis. R package v.3.3.2. [WWW document] URL. https://cran.r-project.org/web/packages/ggplot2/index.html [accessed 17 June 2020], 2016.

Wingett, S. W and Andrews, S.: FASTQ screen: a tool for multi-genome mapping and quality control, F1000Research, 7, 1338, 2018.

Zhang, Y., Li, Z. K., Schulz, K. G., Hu, Y., Irwin, A. J., and Finkel, Z. V.: Growth-dependent changes in elemental stoichiometry and macromolecular allocation in the coccolithophore *Emiliania huxleyi* under different environmental conditions, Limnol. Oceanogr., 66, 2999–3009, doi: 10.1002/lno.11854, 2021.